## Research Article

implementation science; MeHPriC; mhGAP; Nigeria; primary health care; training evaluation

**Corresponding author:**
Abiodun Adewuya;
Email: abiodun.adewuya@lasucom.edu.ng

# Scaling mental health care in Nigeria: Impact of WHO mhGAP training under the MeHPriC program on knowledge, attitudes, and practices of primary health care workers in Lagos State – A pre-post mixed-methods study

Abiodun Adewuya[1,2] ⓘ, Bolanle Ola[1], Olurotimi Coker[1], Olayinka Atilola[1], Olushola Olibamoyo[1], Olabisi Oladipo[2] and Tolu Ajomale[3]

[1]Department of Behavioural Medicine, Lagos State University College of Medicine, Ikeja, Nigeria; [2]Research Unit, Centre for Mental Health Research and Initiative, Ikeja, Nigeria and [3]Mental Health Desk Office, Lagos State Ministry of Health, Ikeja, Nigeria

## Abstract

The growing burden of mental, neurological and substance use (MNS) disorders in low-resource settings has prompted efforts to integrate mental health into primary health care (PHC). This study evaluated the implementation and outcomes of a large-scale mhGAP training initiative under the Mental Health in Primary Care (MeHPriC) program in Lagos State, Nigeria. A total of 852 PHC workers from 57 facilities completed a 5-day mhGAP training and a 1-day refresher session. Using a pre-post mixed-methods design, we assessed changes in knowledge, stigma, clinical practice and self-efficacy, with follow-up at five months. Quantitative findings revealed significant improvements in knowledge and attitudes, with enhanced clinical practice reported by 69.1% of participants. Supervision, knowledge gains and self-efficacy emerged as predictors of improved practice. Qualitative data, analyzed using the Consolidated Framework for Implementation Research (CFIR), highlighted increased confidence, reduced stigma and the enabling role of supervision and peer support, alongside persistent barriers such as medication stock-outs and limited referral networks. The study offers robust evidence for the effectiveness of task-sharing approaches when supported by contextual adaptation and system-level readiness. The MeHPriC model demonstrates that government-led mhGAP scale-up in PHC is both feasible and impactful, offering a replicable pathway for mental health integration in other LMICs.

## Impact statement

This study represents one of the most comprehensive real-world evaluations of the World Health Organization's Mental Health Gap Action Programme (mhGAP) training in sub-Saharan Africa. Conducted within the Mental Health in Primary Care (MeHPriC) program in Lagos State, Nigeria, this pre-post mixed-methods study evaluated the knowledge, attitudes and practice changes among 852 primary health care (PHC) workers trained across 57 Flagship PHC facilities. The findings demonstrate that mhGAP training, when scaled across a decentralized health system, significantly enhances the mental health service capacity of frontline providers. Participants exhibited substantial improvements in diagnostic confidence, reduced stigma and greater willingness to manage mental health conditions, including depression, other significant mental health complaints (including anxiety-related symptoms), psychosis, epilepsy and suicide risk. These outcomes were not merely statistical but reflected profound shifts in perception and practice across diverse cadres, including doctors, nurses, community health officers, pharmacy technicians and social workers/counselors. However, the study underscores that training alone is insufficient. Effective implementation relied on supportive supervision, availability of essential psychotropic medications, peer-to-peer WhatsApp support groups and robust referral systems. By integrating quantitative outcomes with qualitative insights, this study provides an implementation roadmap for other low- and middle-income countries (LMICs) seeking to integrate mental health services into PHC systems. The scale, rigor and mixed-methods design of this study offer timely evidence for national policymakers, global mental health stakeholders and funders. It affirms that scaling up mental health training is not only feasible but transformative when embedded within a supportive health system and informed by the lived experiences of frontline providers.

## Introduction

Mental, neurological and substance use (MNS) disorders constitute a substantial global health challenge, contributing approximately 14% of disability-adjusted life years worldwide, yet receiving less than 1% of health budgets in low- and middle-income countries (LMICs) (GBD, 2022). In Nigeria, a nation of over 200 million people, an estimated 20–30% of the population experiences MNS disorders annually, but fewer than 10% access any form of treatment, and less than 1% receive specialist mental health care (Gureje et al., 2010; Adewuya et al., 2018). This profound treatment gap is exacerbated by a critical shortage of mental health professionals, with only about 300 psychiatrists serving the entire population, yielding a ratio of 1 psychiatrist per 1.5 million people (Fadele et al., 2024). The majority of Nigerians, over 75% of whom reside in rural and peri-urban areas, face additional barriers, including pervasive stigma, limited mental health awareness and inadequate integration of mental health services into primary health care (PHC) systems (Abdulmalik et al., 2013). Addressing this gap necessitates scalable, evidence-based interventions that leverage existing PHC infrastructure through task-sharing, a strategy championed by the World Health Organization's Mental Health Gap Action Programme (mhGAP) to deliver mental health care in non-specialized settings (WHO, 2008; Patel et al., 2010). Central to this initiative is the mhGAP Intervention Guide (mhGAP-IG), an evidence-based clinical tool designed to enable non-specialist health workers to identify, manage and refer individuals with priority MNS disorders, including depression, psychosis, epilepsy, substance use disorders and suicide risk (WHO, 2016). Adopted in over 90 countries and adapted to local contexts, the mhGAP-IG aligns with WHO's global mental health action framework (Keynejad et al., 2021).

Various training models based on the mhGAP Training Manual have emerged, including 5-day intensive workshops, modular cascade training and digital reinforcement sessions (Mills and Lacroix, 2019). These models combine didactic instruction with participatory methods, such as clinical vignettes, role-plays, algorithm navigation and flowchart-based simulations. Evaluations of these training modalities demonstrate improvements in health workers' knowledge and self-efficacy, though evidence of sustained clinical behavior change remains limited (Jordans et al., 2012; Ahrens et al., 2020; Al-Uzri et al., 2024; Humayun and Najmussaqib, 2024).

Nigeria's engagement with mhGAP implementation has evolved significantly over the past decade, with early initiatives providing foundational evidence for task-sharing in PHC (Abdulmalik et al., 2013; Gureje et al., 2015; Eaton et al., 2018; Adewuya et al., 2019; Chu et al., 2022). A seminal effort in Ogun State, evaluated by Adebowale et al. (2014), trained 80 PHC workers (90% female, 87.5% nurses) across 20 local government areas in a 3-day mhGAP course, employing didactic lectures, role-plays and video demonstrations to address five priority conditions: psychosis, depression, alcohol and substance abuse, epilepsy and other significant emotional complaints (OSEC). The study reported significant improvements in diagnostic accuracy and treatment planning, with 473 patients treated over 12 months. The Ogun State program highlighted the feasibility of PHC-based mental health care, emphasizing the critical role of ongoing supportive supervision and algorithm-based decision-making. However, its smaller scale, reliance on vignette-based assessments, absence of long-term follow-up, and lack of qualitative insights into implementation dynamics limited its scope. Other Nigerian initiatives, such as the HAPPINESS project in Imo State (Chu et al., 2022) and the mhSUN program in northern states (Eaton et al., 2018), have similarly demonstrated knowledge and skill improvements among PHC workers but often lack sustained outcome evaluation or implementation science frameworks to elucidate enablers and barriers.

### Policy and system context

In Lagos State, Nigeria's most populous and economically significant state, efforts to scale up mental health services at the primary health care (PHC) level are guided by the Mental Health in Primary Care (MeHPriC) program. Anchored in WHO's mhGAP-IG, MeHPriC operates within a robust policy environment shaped by the Lagos State Mental Health Policy (2015), which explicitly prioritizes the integration of mental health services into PHC through task-sharing and community-based care models. This policy aligns with Nigeria's National Mental Health Policy (2013) and is supported by strong political will, evidenced by sustained government funding, advocacy from the Lagos State Ministry of Health and the Office of the Governor, and strategic partnerships with academic institutions and non-governmental organizations. A pilot study of MeHPriC, implemented across 10 PHC facilities in Lagos, demonstrated the feasibility, acceptability, and initial clinical impact of mhGAP-guided service delivery (Adewuya et al., 2019). The policy framework has facilitated the scale-up of mhGAP training to 57 Flagship PHCs, selected for their enhanced infrastructure and staffing capacity. System strengths include a well-established PHC network, a trained and diverse workforce, and digital infrastructure, such as WhatsApp-based peer support groups, which enable real-time communication and resource sharing among PHC workers. However, significant challenges persist, including recurrent psychotropic medication stock-outs, limited specialist referral pathways due to the scarcity of psychiatrists, and high staff workloads driven by competing health priorities, such as maternal and child health services. These constraints echo resource limitations observed in Ogun State (Adebowale et al., 2014), where medication availability and referral access similarly hindered implementation. Addressing these systemic barriers requires health system strengthening, including improved supply chains, enhanced referral networks, and workload management, to sustain mhGAP integration and ensure equitable mental health care delivery.

The MeHPriC program was designed to bridge Nigeria's mental health treatment gap by training PHC workers in Lagos State to deliver mhGAP-based care for five priority conditions selected based on local epidemiological data, reflecting high prevalence, service burden in PHC settings and stakeholder consensus. This study evaluates the impact of a 5-day mhGAP training, complemented by a 1-day refresher session, on PHC workers' knowledge, attitudes and practices, using a pre-post mixed-methods design with a 5-month follow-up. By integrating validated quantitative scales with qualitative insights, collected through culturally adapted English and Yoruba guides, the study examines training effectiveness alongside implementation enablers (e.g., supervision, peer support) and barriers (e.g., medication stock-outs, stigma). The findings aim to inform policy and practice for scaling mhGAP in Nigeria and other LMICs, addressing critical gaps in sustained outcome evaluation, implementation science and cultural adaptation identified in prior Nigerian efforts.

## Methodology

### Study design

This study employed a pre-post mixed-methods design to evaluate the impact of a 5-day mhGAP training program, with a 1-day

refresher session, on the knowledge, attitudes, and practices of PHC workers in Lagos State, Nigeria, under the MeHPriC initiative. Quantitative data were collected at baseline (pre-training), immediately post-training, and at 5-month follow-up to assess changes in outcomes. Qualitative data, collected at 5 months post-training through focus group discussions (FGDs) and key informant interviews (KIIs), explored implementation experiences and contextual factors influencing mhGAP uptake. The study was conducted between January and June 2021 across 57 Flagship PHCs in Lagos State, selected for their enhanced infrastructure and staffing capacity.

### Participant selection

Participants comprised 852 PHC workers from 57 Flagship PHCs across urban and rural Local Council Development Areas (LCDAs) in Lagos State, reflecting the diversity of Nigeria's PHC workforce. Nominations were made by PHC facility managers in collaboration with LCDA PHC coordinators, based on criteria ensuring representation across cadres (doctors, nurses/midwives, community health officers [CHOs], pharmacy technicians, social workers/counselors, and clerical staff) and perceived commitment to service delivery. The selection process prioritized inclusivity to support team-based mental health care, involving roles in screening, diagnosis, management, documentation and referral. No formal assessment of participants' interest or future availability for mental health roles was conducted due to logistical constraints, relying instead on managers' informal evaluations of dedication and prior engagement in community health or non-communicable disease programs. This limitation is acknowledged, as motivation assessments could enhance training effectiveness. The sample's demographic profile aligns with Lagos State's PHC workforce ensuring representativeness.

### Training design and delivery

The training curriculum, adapted from the WHO mhGAP-IG (WHO, 2016), targeted five priority conditions selected based on local epidemiology, service burden and stakeholder consensus: depression, other significant mental health complaints (including anxiety-related symptoms), psychosis, epilepsy and suicide risk. The inclusion of anxiety-related symptoms under "other significant mental health complaints" was tailored to address their high prevalence in Lagos State, as permitted by WHO adaptation guidelines. Delivered over 5 days, with a 1-day refresher session at 3 months, the training was facilitated by mental health specialists from the Lagos State Ministry of Health and academic partners, using didactic lectures, interactive role-plays, video demonstrations and case-based discussions, guided by the mhGAP Training Manual (WHO, 2017).

Training content was tailored to cadre-specific roles to ensure ethical alignment with professional scopes, guided by the mhGAP Operations Manual (2018). Clinical cadres (doctors, nurses, CHOs) received comprehensive training on diagnostic algorithms, treatment protocols (e.g., psychoeducation, psychotropic prescribing) and clinical decision-making for the five conditions. Non-clinical cadres (pharmacy technicians, social workers/counselors, and clerical staff) focused on supportive tasks, including identification of mental health issues, basic psychoeducation, documentation using mhGAP flowcharts, referral facilitation to specialists and creating therapeutic environments. Role-plays simulated PHC scenarios, such as screening for suicide risk in busy clinics or counseling epilepsy patients to reduce stigma, and ensuring practical application. This differentiation, respected professional boundaries while

fostering team-based care. Training was conducted in batches of approximately 170 participants across urban and rural LCDAs, with materials translated into Yoruba where needed to enhance accessibility.

### Supervision and implementation support

Post-training supervision was structured to reinforce skills and support mhGAP implementation, integrating with routine PHC systems for sustainability. During the initial 6–12 months, each PHC received monthly 1–2-h supervisory visits from mental health trainers, following standardized checklists covering: (1) review of cases managed using mhGAP protocols, (2) assessment of algorithm adherence, (3) problem-solving for complex presentations, (4) skill reinforcement through direct observation, (5) documentation and referral practice evaluation, and (6) addressing implementation barriers (e.g., medication stock-outs). Supervisors collaborated with LCDA PHC coordinators to align visits with existing oversight mechanisms, ensuring integration.

WhatsApp peer support groups, established for each training batch (approximately 170 members per group), facilitated ongoing learning and problem-solving. Moderated by master trainers, these groups enabled weekly case discussions, troubleshooting (e.g., managing psychosis in resource-limited settings) and sharing of updated mhGAP resources, with an average of 10–15 messages per week per group. This digital infrastructure leveraged Lagos State's robust connectivity, enhancing implementation fidelity. Supervision and peer support were critical enablers, as evidenced by qualitative data (Supplementary Table S1).

### Measurement instruments

Quantitative data were collected using four adapted instruments, pilot-tested with 30 PHC workers to ensure cultural and linguistic appropriateness, with adjustments to reduce stigma (e.g., "mental health conditions" instead of "mental illness"). The adaptation process involved cognitive interviews, expert reviews by Nigerian mental health specialists and PHC managers, and Yoruba translations validated by local linguists, ensuring contextual relevance while preserving psychometric properties. Reliability was assessed via Cronbach's alpha. The full instruments, including scoring and adaptation notes are in Supplementary File S1.

1. **mhGAP knowledge questionnaire**: A 25-item multiple-choice tool, adapted from the mhGAP training manuals (WHO, 2017), assessed knowledge of the five priority conditions (e.g., appropriate interventions for depression). Scores ranged from 0 to 25, categorized as low (0–10), moderate (11–20), or high ($\geq$21), with $\alpha = 0.82$.
2. **Stigma and attitude scale**: A 15-item Likert-scale (1 = strongly disagree, 5 = strongly agree), adapted from mhGAP training manuals (WHO, 2017) and the mental illness: Clinicians' Attitudes (MICA) Scale (Gabbidon et al., 2013), measured beliefs about MNS disorders (e.g., "people with psychosis are dangerous"). Scores ranged from 15 to 75, with higher scores indicating greater stigma (low: 15–30, moderate: 31–54, high: $\geq$55), with $\alpha = 0.78$.
3. **Self-reported practice change survey**: A 10-item Likert-scale (1 = never, 5 = always), adapted from mhGAP training manual (WHO, 2017) and Kirkpatrick's Level 3 Evaluation (Kirkpatrick and Kirkpatrick, 2006), assessed frequency of mhGAP-aligned behaviors (e.g., suicide risk screening). Scores ranged from 10 to

50, categorized as low (10–24), moderate (25–34), or enhanced (≥35), with the threshold of ≥35 established through pilot-testing to reflect consistent engagement (average item score ≥ 3.5) and validated by expert consultation, with $\alpha = 0.85$.

4. **Self-efficacy scale**: A 10-item Likert-scale (1 = not at all confident, 5 = extremely confident), adapted from Bandura's Self Efficacy Scale (2006), assessed confidence in applying mhGAP protocols (e.g., diagnosing depression). Scores ranged from 10 to 50, with $\alpha = 0.80$.

The reliance on self-reported measures, particularly for practice and stigma, introduces a risk of social desirability bias, potentially overestimating outcomes. This was mitigated through anonymous data collection, triangulation with qualitative findings and explicit acknowledgment of limitations. Alternative measures, such as direct observation or patient audits, were infeasible due to resource constraints and incomplete PHC documentation.

### Qualitative data collection

Qualitative data were collected at 5 months post-training through 12 FGDs (6–8 participants each, $n = 84$, purposively sampled from urban/rural LCDAs across cadres) and 10 KIIs with strategic actors (5 LCDA PHC coordinators, 3 master trainers, 2 Lagos State Ministry of Health officials). FGDs explored: (1) perceived training impact, (2) confidence in managing MNS disorders, (3) organizational barriers, (4) stigma changes, (5) use of job aids/WhatsApp groups/supervision, and (6) sustainability suggestions. KIIs addressed: (1) training feasibility, (2) PHC system integration, (3) LCDA/facility/Ministry coordination, and (4) supervision practices. Guides, co-developed in English and Yoruba for cultural accessibility, were structured around Consolidated Framework for Implementation Research (CFIR) domains (Damschroder et al., 2009) and are provided in Supplementary File S2. Audio-recorded sessions, conducted in private PHC settings, were transcribed verbatim, with Yoruba transcripts translated into English by bilingual researchers and validated for accuracy.

### Data analysis

#### Quantitative analysis

Quantitative data were analyzed using SPSS Version 26. Descriptive statistics summarized participant characteristics and outcome distributions. Pre-post changes in knowledge and stigma were assessed using paired $t$-tests for continuous scores and Cochran's $Q$ tests for categorical shifts, with McNemar's tests for pairwise comparisons. Effect sizes (Cohen's $d$) were calculated to quantify change magnitude (e.g., $d = 0.97$ for knowledge post-training). Practice change and self-efficacy, measured at 5 months, were analyzed descriptively, with logistic regression models estimating predictors of enhanced practice (score ≥ 35), controlling for cadre, age, experience and supervision exposure. Missing data (2.9%, $n = 25$ at follow-up, primarily due to staff transfers) were handled using listwise deletion, with sensitivity analyses confirming minimal bias. All analyses used a significance level of $p < 0.05$, with 95% confidence intervals reported.

### Qualitative analysis

Thematic analysis, guided by CFIR (Braun and Clarke, 2006; Damschroder et al., 2009), was conducted by two independent analysts using NVivo 12. Coding covered CFIR domains: Characteristics of the Intervention (e.g., training adaptability), Inner Setting (e.g., resource availability), Characteristics of individuals (e.g., self-efficacy), and implementation process (e.g., supervision effectiveness). An initial codebook was developed deductively from CFIR, with inductive codes added for emergent themes (e.g., desire for mentorship). Inter-rater reliability was assessed (kappa = 0.82), with discrepancies resolved through consensus. Five themes were identified, mapped to CFIR domains, and illustrated with quotes (Supplementary Table S1). A mixed-methods joint display integrated quantitative (e.g., 69.1% enhanced practice) and qualitative findings (e.g., confidence narratives) to show convergence.

## Results

### Participant characteristics

Of the 852 PHC workers trained across 57 Flagship PHCs in Lagos State, all completed baseline and post-training assessments, with 827 (97.1%) successfully followed up at 5 months. The sample was predominantly female (66.9%, $n = 570$), reflecting the gender distribution of Lagos State's PHC workforce (68% female). Nurses/midwives comprised 43.5% ($n = 371$), followed by community health officers (CHOs; 24.8%, $n = 211$), doctors (9.4%, $n = 80$), pharmacy technicians (8.9%, $n = 76$), social workers/counselors (7.3%, $n = 62$) and clerical staff (6.1%, $n = 52$), with a slight overrepresentation of nurses (43.5% vs. ~40% in the workforce) controlled for in regression analyses to minimize bias. The mean age was 39.2 years (SD = 8.7), with 42.3% ($n = 360$) having over 10 years of experience (range: 1–30 years). Participants were distributed across urban (58.7%, $n = 500$) and rural (41.3%, $n = 352$) LCDAs, ensuring geographic diversity. Missing data at follow-up (2.9%, $n = 25$), primarily due to staff transfers, were handled using listwise deletion, with sensitivity analyses confirming no significant bias in outcomes by cadre, age, or location. Table 1 provides a detailed breakdown by cadre.

### Knowledge outcomes

The mhGAP training significantly improved PHC workers' knowledge of the five priority conditions. Mean scores on the 25-item mhGAP Knowledge Questionnaire increased from 14.81 (SD = 6.53) at baseline to 20.25 (SD = 4.68) post-training ($t = -28.29$, $p < 0.001$, $d = 0.97$, 95% CI $[-5.88, -5.00]$), slightly declining to 19.49 (SD = 5.33) at 5-month follow-up but remaining significantly higher than baseline ($t = -21.12$, $p < 0.001$, $d = 0.72$, 95% CI $[-5.11, -4.25]$). Cochran's $Q$ test confirmed significant shifts in knowledge categories over time ($Q = 412.6$, $p < 0.001$), with McNemar's tests showing the proportion of participants with "High" knowledge (score ≥ 21) rising from 19.8% ($n = 169$) at baseline to 62.4% ($n = 531$) post-training ($p < 0.001$) and sustained at 56.6% ($n = 468$) at follow-up ($p < 0.001$). Disorder-specific analysis revealed the greatest improvements for suicide risk (mean increase = 2.8 points, $d = 0.85$) and other significant mental health complaints (mean increase = 2.6 points, $d = 0.78$), which were poorly understood at baseline (mean scores <3.0), followed by depression ($d = 0.65$), psychosis ($d = 0.60$), and epilepsy ($d = 0.55$).

### Stigma and attitude outcomes

The training significantly reduced measured mental health stigma, though findings should be interpreted cautiously due to potential social desirability bias. As shown in Table 2, mean scores on the 15-item Stigma and Attitude Scale decreased from 49.46 (SD = 15.63)

**Table 1.** Participant characteristics of PHC workers in the MeHPriC program (*N* = 852)

| Characteristic | Frequency | Percentage (%) | Mean (SD) | Notes |
|---|---|---|---|---|
| **Gender** | | | | |
| Female | 570 | 66.9 | | Aligns with Lagos State PHC workforce (68% female; Lagos State Primary Health Care Board, 2017) and Adebowale et al. (2014) (90% female). |
| Male | 282 | 33.1 | | |
| **Cadre** | | | | |
| Nurses/ midwives | 371 | 43.5 | | Slight overrepresentation (43.5% vs. ~40% in workforce), controlled in analyses. |
| Community Health Officers (CHOs) | 211 | 24.8 | | |
| Doctors | 80 | 9.4 | | |
| Pharmacy technicians | 76 | 8.9 | | |
| Social Workers/ counselors | 62 | 7.3 | | |
| Clerical staff | 52 | 6.1 | | |
| **Age (years)** | | | 39.2 (8.7) | Range: 22–60 years. |
| **Experience (years)** | | | 12.4 (7.9) | 42.3% (*n* = 360) with >10 years, range: 1–30 years. |
| **Location** | | | | |
| Urban LCDAs | 500 | 58.7 | | Ensures geographic diversity. |
| Rural LCDAs | 352 | 41.3 | | |

at baseline to 38.53 (SD = 16.30) post-training (*t* = 14.04, *p* < 0.001, *d* = 0.48, 95% CI [8.75, 11.79]), stabilizing at 39.19 (SD = 16.19) at 5-month follow-up (*t* = 13.21, *p* < 0.001, *d* = 0.46, 95% CI [8.45, 11.09]). Cochran's *Q* test indicated significant changes in stigma categories (*Q* = 287.4, *p* < 0.001), with McNemar's tests confirming a reduction in "High" stigma (score ≥ 55) from 42.0% (*n* = 358) at baseline to 19.5% (*n* = 167) post-training (*p* < 0.001) and 19.5% (*n* = 161) at follow-up (*p* < 0.001). Reductions were most pronounced for attitudes toward psychosis (mean decrease = 1.8 points, *d* = 0.52) and epilepsy (mean decrease = 1.6 points, *d* = 0.49), reflecting training emphasis on medical explanations, compared to smaller changes for depression (*d* = 0.35) and suicide risk (*d* = 0.30). These findings align with qualitative accounts of increased empathy (Supplementary Table S1).

## Practice change outcomes

At 5-month follow-up, 69.1% of participants (*n* = 571) demonstrated "Enhanced Practice" on the 10-item Self-Reported Practice Change Survey, defined as a score ≥ 35 (mean = 37.05, SD = 8.45),

established through pilot-testing with 30 PHC workers to reflect consistent engagement (average item score ≥ 3.5, indicating "Often" or "Always") and validated by expert consultation (WHO, 2018). Frequently reported behaviors included screening for suicide risk (82.3%, *n* = 681), using mhGAP flowcharts for diagnosis (76.5%, *n* = 633), and referring complex cases (74.8%, *n* = 619). Practice change was higher among clinical cadres (doctors: 78.8%, *n* = 63; nurses: 73.0%, *n* = 271) than non-clinical cadres (pharmacy technicians: 62.7%, *n* = 47; clerical staff: 55.8%, *n* = 29), reflecting role-specific training.

## Self-efficacy outcomes

Self-efficacy in applying mhGAP protocols was high at 5-month follow-up, with a mean score of 42.15 (SD = 7.22) on the 10-item Self-Efficacy Scale, indicating strong confidence (average item score ≥ 4.2). Highest confidence was reported for diagnosing depression (88.4%, *n* = 731) and epilepsy (85.7%, *n* = 709), followed by suicide risk assessment (81.2%, *n* = 672) and managing psychosis (78.5%, *n* = 649). Confidence was lower for other significant mental health complaints (70.3%, *n* = 581), reflecting complexity in anxiety-related presentations. Clinical cadres reported higher self-efficacy (doctors: mean = 45.3, SD = 5.8; nurses: mean = 43.2, SD = 6.9) than non-clinical cadres (clerical staff: mean = 38.7, SD = 8.1), consistent with training focus. These findings complement qualitative narratives of increased confidence (Supplementary Table S1).

## Predictors of enhanced practice

Logistic regression, controlling for cadre, age, experience, and supervision exposure, identified key predictors of enhanced practice (score ≥ 35) at 5-month follow-up. Being a doctor (adjusted OR = 2.3, 95% CI [1.5, 3.6], *p* < 0.001) or nurse (adjusted OR = 1.7, 95% CI [1.2, 2.4], *p* = 0.003) was significantly associated with enhanced practice compared to non-clinical cadres, reflecting clinical authority and training emphasis. As shown in Table 3, monthly supervision visits (adjusted OR = 4.3, 95% CI [2.8, 6.5], *p* < 0.001) and active WhatsApp group participation (adjusted OR = 2.1, 95% CI [1.4, 3.2], *p* = 0.001) were strong predictors, underscoring support mechanisms' role. Age and experience were non-significant, suggesting training accessibility across demographics. These findings align with qualitative data on supervision's impact (Supplementary Table S1).

## Qualitative findings

Thematic analysis of 12 FGDs and 10 KIIs, guided by the CFIR, yielded five themes mapped to CFIR domains (see Figure 1). Co-developed English and Yoruba guides (Supplementary File S2) ensured cultural accessibility, enriching data richness. Additional quotes are provided in Supplementary Table S1.

1. **Increased clinical confidence (individual characteristics)**: Participants reported enhanced diagnostic and communication skills, particularly for suicide risk and epilepsy. Sub-themes included confidence in using mhGAP flowcharts and engaging patients empathetically. A nurse (rural PHC) stated, "*Before, I was scared to ask about suicide. Now, I can ask directly and know what to do.*"
2. **Stigma reduction and empathy (individual characteristics)**: Training fostered empathy, especially for psychosis and

**Table 2.** Knowledge, stigma and practice change outcomes across timepoints

| Outcome Measure | Baseline (*n* = 852) | Post-Training (*n* = 852) | 5-Month Follow-Up (*n* = 827) | Statistical Tests |
|---|---|---|---|---|
| Knowledge outcomes | | | | |
| Mean score (SD) | 14.81 (6.53) | 20.25 (4.68) | 19.49 (5.33) | Baseline to Post-Training: *t* = −28.29, *p* < 0.001, *d* = 0.97, 95% CI [−5.88, −5.00] Baseline to Follow-Up: *t* = −21.12, *p* < 0.001, *d* = 0.72, 95% CI [−5.11, −4.25] |
| Categorical knowledge levels | | | | Cochran's *Q* = 412.6, *p* < 0.001 McNemar's: *p* < 0.001 (baseline vs. follow-up) |
| Low knowledge (<15) | 356 (41.8%) | 71 (8.3%) | 105 (12.7%) | |
| Moderate knowledge (15–20) | 327 (38.4%) | 250 (29.3%) | 254 (30.7%) | |
| High knowledge (≥21) | 169 (19.8%) | 531 (62.4%) | 468 (56.6%) | |
| Disorder-specific effect sizes | | | | |
| Suicide risk | | | *d* = 0.85 | |
| Other significant mental health complaints | | | *d* = 0.78 | |
| Depression | | | *d* = 0.65 | |
| Psychosis | | | *d* = 0.60 | |
| Epilepsy | | | *d* = 0.55 | |
| Stigma outcomes | | | | |
| Mean score (SD) | 49.46 (15.63) | 38.53 (16.30) | 39.19 (16.19) | Baseline to Post-Training: *t* = 14.04, *p* < 0.001, *d* = 0.48, 95% CI [8.75, 11.79] Baseline to Follow-Up: *t* = 13.21, *p* < 0.001, *d* = 0.46, 95% CI [8.45, 11.09] |
| Categorical stigma levels | | | | Cochran's *Q* = 287.4, *p* < 0.001 McNemar's: *p* < 0.001 (baseline vs. follow-up) |
| Low stigma (15–34) | 150 (17.6%) | 422 (49.6%) | 378 (45.7%) | |
| Moderate stigma (35–54) | 344 (40.4%) | 263 (30.9%) | 288 (34.8%) | |
| High stigma (≥55) | 358 (42.0%) | 167 (19.5%) | 161 (19.5%) | |
| Condition-specific Stigma reduction | | | | |
| Psychosis | | | *d* = 0.50 | |
| Epilepsy | | | *d* = 0.47 | |
| Depression | | | *d* = 0.35 | |
| Suicide risk | | | *d* = 0.30 | |
| Practice change outcomes | | | | |
| Mean score (SD) | – | – | 37.05 (8.45) | |
| Enhanced practice (≥35) | – | – | 571 (69.1%) | Threshold validated through pilot-testing |
| Specific practice behaviors | | | | |
| Mental health screening | – | – | 681 (82.3%) | "Often" or "Always" |
| Suicide risk assessment | – | – | 672 (81.2%) | |
| mhGAP flowchart use | – | – | 633 (76.5%) | |
| Complex case referral | – | – | 619 (74.8%) | |
| Patient/family Psychoeducation | – | – | 589 (71.2%) | |
| Practice by cadre | | | | |
| Doctors | – | – | 63/80 (78.8%) | |
| Nurses/midwives | – | – | 271/371 (73.0%) | |
| Community health officers | – | – | 123/211 (58.3%) | |
| Pharmacy technicians | – | – | 48/76 (63.2%) | |

(*Continued*)

**Table 2.** (*Continued*)

| Outcome Measure | Baseline (*n* = 852) | Post-Training (*n* = 852) | 5-Month Follow-Up (*n* = 827) | Statistical Tests |
|---|---|---|---|---|
| Social workers/counselors | – | – | 38/62 (61.3%) | |
| Clerical staff | – | – | 29/52 (55.8%) | |

*Notes*:
- Knowledge (25-item mhGAP Knowledge Questionnaire, $\alpha$ = 0.82), Stigma (15-item Stigma and Attitude Scale, $\alpha$ = 0.78) and Practice (10-item Self-Reported Practice Change Survey, $\alpha$ = 0.85) assessed via validated tools.
- Other significant mental health complaints include anxiety-related symptoms, per mhGAP-IG (2016).
- Effect sizes calculated using Cohen's *d*.
- Cochran's *Q* and McNemar's tests assessed categorical changes.
- Enhanced practice threshold (≥35) pilot-tested with 30 PHC workers, validated by expert consultation.
- Missing data (2.9%, *n* = 25, due to staff transfers) handled via listwise deletion; sensitivity analyses confirmed minimal bias.

epilepsy, reducing avoidance behaviors. A CHO (urban PHC) noted, "*We used to avoid epileptics. Now we understand it's a medical condition and they deserve respect.*" Persistent cultural beliefs were acknowledged, aligning with quantitative stigma findings.

3. **System-level challenges (inner setting)**: Barriers included medication stock-outs (e.g., antipsychotics unavailable in 60% of PHCs), lack of private consultation spaces and high workloads. A doctor (peri-urban PHC) said, "*Flowcharts help, but we lack drugs and space to counsel privately,.*"

4. **Role of supervision and peer support (process)**: Monthly supervision and WhatsApp groups were critical enablers. A midwife (PHC) reported, "*Supervisors reviewed cases, and our WhatsApp group helped troubleshoot complex psychosis cases.*" Groups averaged 10–15 weekly messages, enhancing fidelity.

5. **Desire for expansion and mentorship (outer setting)**: Participants advocated for child mental health modules and mentorship roles. A CHO (rural PHC) said, "*We want mhGAP for children and to mentor others in our LCDA,*" which reflects enthusiasm for scaling.

Figure 1 is a thematic map illustrating interconnections between qualitative themes, aligned with CFIR domains.

### Mixed-methods integration

A mixed-methods joint display analysis revealed strong convergence between quantitative and qualitative findings, enhancing interpretive robustness. Quantitative knowledge gains (Cohen's *d* = 0.72) aligned with qualitative narratives of increased clinical confidence, particularly for suicide risk and epilepsy. Measured stigma reduction (*d* = 0.46) was corroborated by accounts of empathy toward psychosis and epilepsy patients, though qualitative data noted persistent beliefs (Supplementary Table S1). Enhanced practice (69.1%) converged with reports of flowchart use and supervision benefits, while system-level barriers (e.g., stock-outs) explained practice variability.

### Discussion

#### Summary of key findings

The MeHPriC program's training significantly enhances PHC workers' capacity to deliver mental health services in Nigeria's resource-constrained context. Training 852 workers across 57 Flagship PHCs improved knowledge, reduced stigma and increased clinical engagement for depression, other significant mental health complaints (including anxiety-related symptoms), psychosis, epilepsy and suicide risk. Qualitative findings highlight heightened confidence and empathy, with supervision and WhatsApp peer groups as critical enablers. System-level barriers, such as medication stock-outs in 60% of PHCs, constrained implementation, underscoring the need for health system strengthening. This study offers a scalable model for sub-Saharan Africa, aligning with global task-sharing efforts to address mental health treatment gaps.

### Changes in knowledge, attitude/stigma, and practice enhancement

The mhGAP training significantly enhanced PHC workers' knowledge, attitudes and practices for managing MNS disorders. Knowledge gains were sustained over five months, with notable improvements in suicide risk and other significant mental health complaints, reflecting the training's focus on high-priority areas with low baseline understanding. Compared to Adebowale et al. (2014), which reported 12.5% diagnostic accuracy improvements for psychosis and 30% for other significant emotional complaints among 80 PHC workers in Ogun State, our study's larger cohort and validated scales demonstrate broader and sustained impact. Gureje et al. (2015) earlier noted 40,8% knowledge gains but lacked longitudinal follow-up. Globally, Ayano et al. (2017) reported a 101% knowledge increase in Ethiopia, while Bellizzi et al. (2021) noted a 102% knowledge/attitude increase in Egypt, suggesting Nigeria's gains align with global trends but required cultural adaptations like Yoruba-language focus group discussion (FGD) guides. Stigma reduction, particularly for psychosis and epilepsy, was a novel contribution, though cultural sensitivities require cautious interpretation. While Adebowale et al. (2014) did not assess stigma, Chu et al. (2022) in Imo State reported qualitative reductions without quantitative measures. Ayano et al. (2017) noted a 99% attitude improvement in Ethiopia, but Nigeria's context, with persistent spiritual beliefs, necessitated tailored training to address stigma effectively. Enhanced practice, including frequent screening and mhGAP flowchart use, was achieved by a majority of participants, driven by clinical cadres like doctors and nurses. This surpasses Adebowale et al. (2014)'s vignette-based outcomes, where treatment intervention scores improved 77.9–114%. Qualitative narratives of improved patient interactions reinforce these gains, positioning task-sharing as a viable strategy for LMICs.

### Predictors of enhanced practice

This study is among the few to identify statistically significant predictors of sustained practice change, providing critical insights

**Table 3.** Logistic regression predictors of enhanced practice (score ≥ 35) at 5-month follow-up

| Predictor variable | Enhanced practice n (%) | Crude OR (95% CI) | Adjusted OR (95% CI) | p-value | Clinical Interpretation |
|---|---|---|---|---|---|
| **Supervision and support** | | | | | |
| Clinical supervision (≥1 visit) | 487/682 (71.4%) | 4.8 (3.2–7.1) | 4.3 (2.8–6.5) | <0.001 | Supervision reinforces protocol adherence and supports complex case management. |
| No supervision | 84/145 (57.9%) | Reference | Reference | | |
| WhatsApp group participation | 356/477 (74.6%) | 2.4 (1.7–3.4) | 2.1 (1.4–3.2) | 0.001 | Digital peer support enhances real-time troubleshooting and fidelity. |
| No WhatsApp participation | 215/350 (61.4%) | Reference | Reference | | |
| **Training outcomes** | | | | | |
| Post-training Knowledge (high ≥21) | 387/468 (82.7%) | 3.2 (2.4–4.3) | 2.8 (2.0–3.9) | <0.001 | High knowledge improves diagnostic accuracy and mhGAP flowchart use. |
| Post-training Knowledge (low/moderate) | 184/359 (51.3%) | Reference | Reference | | |
| Self-efficacy score (high/very high) | – | 1.08 (1.05–1.11) | 1.5 (1.2–1.7) | <0.001 | Confidence drives sustained application of mhGAP skills, especially screening. |
| **Professional cadre** | | | | | |
| Medical officers (doctors) | 63/80 (78.8%) | 3.1 (1.8–5.3) | 2.3 (1.5–3.6) | <0.001 | Doctors lead in adopting mhGAP protocols due to clinical authority. |
| Nurses/midwives | 271/371 (73.0%) | 2.2 (1.6–3.0) | 1.7 (1.2–2.4) | 0.003 | Nurses support task-sharing in resource-limited PHCs. |
| Community health officers | 189/325 (58.1%) | 1.1 (0.8–1.5) | 1.2 (0.8–1.7) | 0.456 | Limited impact reflects less diagnostic authority in PHC settings. |
| Non-clinical cadres* | 48/94 (51.1%) | Reference | Reference | | Supportive roles enhance referrals and documentation. |
| **Demographic factors** | | | | | |
| Gender (female vs. male) | 392/570 (68.8%) vs. 179/257 (69.6%) | 0.96 (0.7–1.3) | 1.1 (0.8–1.5) | 0.672 | Training is accessible across genders, supporting inclusivity. |
| Age (≥40 vs. <40 years) | 289/401 (72.1%) vs. 282/426 (66.2%) | 1.3 (1.0–1.7) | 1.2 (0.9–1.6) | 0.234 | Age does not significantly influence practice change. |
| Experience (≥10 vs. <10 years) | 251/360 (69.7%) vs. 320/467 (68.5%) | 1.1 (0.8–1.4) | 0.9 (0.7–1.2) | 0.512 | Experience has minimal impact, indicating broad applicability. |
| **Contextual factors** | | | | | |
| mhGAP tools available | 541/773 (70.0%) | 1.4 (0.8–2.4) | 1.3 (0.7–2.2) | 0.378 | Tool availability supports but does not significantly drive practice. |
| Tools not available | 30/54 (55.6%) | Reference | Reference | | |
| Prior mental health training | 134/186 (72.0%) | 1.2 (0.8–1.7) | 1.1 (0.8–1.6) | 0.543 | Prior training does not significantly enhance practice change. |
| No prior training | 437/641 (68.2%) | Reference | Reference | | |
| Urban vs. rural location | 339/483 (70.2%) vs. 232/344 (67.4%) | 1.1 (0.9–1.5) | 1.0 (0.7–1.4) | 0.821 | Location does not significantly affect implementation. |

*Notes*:
- Logistic regression model (*n* = 827 complete cases) controlled for all variables.
- Enhanced practice defined as score ≥ 35 on the Self-Reported Practice Change Survey ($\alpha$ = 0.85).
- Missing data (2.9%, *n* = 25, due to staff transfers) handled via listwise deletion; sensitivity analyses confirmed minimal bias.
- Nagelkerke $R^2$ = 0.34, Hosmer–Lemeshow $\chi^2$ = 8.2, *p* = 0.412, area under ROC curve = 0.78 (95% CI: 0.74–0.82) indicate good model fit.
- Bold *p*-values indicate significance (*p* < 0.05).
- *Non-clinical cadres include pharmacy technicians, social workers/counselors, and clerical staff.

for implementation design. Supervision was the strongest predictor, aligning with Ahrens et al. (2020), where frequent supervisory visits in Malawi increased case detection. WhatsApp peer groups, averaging 10–15 weekly messages, enhanced implementation fidelity. High post-training knowledge predicted practice change, consistent with Ayano et al. (2017), where knowledge gains improved psychosis diagnosis. Self-efficacy also drove practice, as in Spagnolo et al.

(2020)'s Tunisia findings, where confidence reduced unnecessary referrals. Clinical cadres (doctors, nurses) showed greater adoption, just like with Adebowale et al. (2014), where clinical roles drove protocol use, while non-clinical cadres supported referrals. Role-play was not significant, contrasting with Ahrens et al. (2020), possibly due to inconsistent supervision, suggesting a need for structured feedback to enhance pedagogical value.

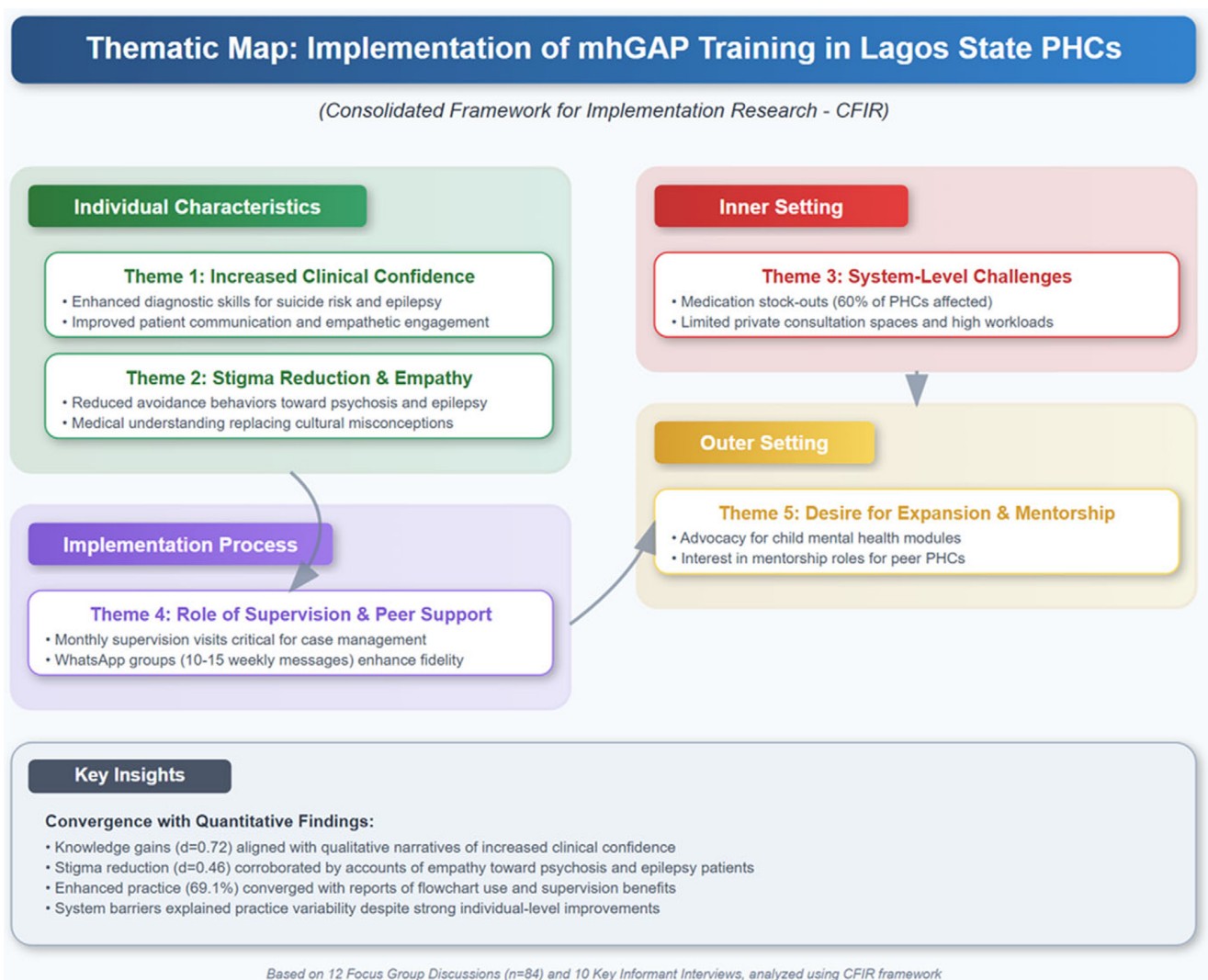

**Figure 1.** Thematic map of qualitative findings.

## Clinical and policy implications

The findings suggest PHC providers can effectively screen and manage MNS disorders with appropriate training, and this supports Nigeria's task-sharing objectives. Stigma reduction, particularly for psychosis and epilepsy, is feasible but requires ongoing cultural sensitivity, as noted in qualitative data. Our Lagos model, featuring multiprofessional cohorts, structured supervision, and digital peer networks, offers a blueprint for national mental health reform. Policy actions should prioritize supply chain reforms to address medication stock-outs in 60% of PHCs, which undermines practice adoption. Strengthening referral networks via telepsychiatry or specialist hubs, would address gaps noted by participants. Integrating digital caseload registers into PHC health management information systems (HMIS) would enable objective monitoring, overcoming the study's limitation and aligning with Adebowale et al. (2014)'s patient tracking. Supervision and WhatsApp groups were pivotal enablers, suggesting investments in digital infrastructure and trainer capacity to sustain implementation. Role-specific training, with non-clinical cadres enhancing referrals and documentation, offers a scalable task-sharing model for resource-constrained settings. Participants' enthusiasm for expanding modules to include child mental health and substance use, alongside mentorship roles, indicates readiness for further scale-up, as evidenced in qualitative accounts. Cultural adaptations, such as Yoruba guides and stigma-sensitive terminology, were instrumental in enhancing rural uptake, suggesting broader investment in localized training materials to reduce stigma and improve service delivery.

## Limitations

Several limitations warrant cautious interpretation of the findings. The absence of objective service delivery measures, such as case-loads by diagnostic category, limits assessment of clinical impact, as Lagos State's PHC HMIS lacked standardized mental health indicators. Qualitative triangulation, documenting frequent screening and referrals, mitigates this gap, but highlights the need for digital HMIS integration to enable robust monitoring. Self-reported measures for stigma and practice may overestimate changes due to social desirability bias, particularly in Nigeria's cultural context, where spiritual beliefs about mental health persist. Anonymous data collection and Yoruba-adapted tools reduce this risk, but observational methods, such as the Enhancing Assessment of Common Therapeutic factors

(ENACT) tool (Kohrt et al., 2015), would enhance validity, though they are resource-intensive in high-workload PHCs. The focus on Flagship PHCs, with potentially higher staffing and infrastructure compared to non-Flagship facilities, limits generalizability to Nigeria's broader PHC network. The absence of formal motivation assessments during participant selection, relying instead on PHC managers' informal evaluations of commitment, may have influenced training uptake, as motivated learners are more likely to apply skills (Keynejad et al., 2021). The lack of a control group limits causal inference, though the large sample size, mixed-methods triangulation, and CFIR-guided qualitative analysis enhance the credibility of the findings.

### Conclusion

This large-scale, mixed-methods evaluation provides robust evidence that the WHO mhGAP-IG, when implemented with fidelity, contextual adaptation and strategic reinforcement, significantly enhances PHC workers' capacity to deliver mental health services in Nigerian settings. Conducted across 57 Flagship PHCs with 852 workers from diverse cadres, the study demonstrates sustained knowledge gains, measured stigma reduction and enhanced clinical practice, driven by supervision and digital peer networks. Qualitative convergence, highlighting increased confidence and empathy amidst structural constraints like medication stock-outs, underscores the training's impact. Compared to prior Nigerian studies in Ogun and Imo States, and to global evaluations in Ethiopia and Egypt, this study's scale, 5-month follow-up, and CFIR-guided qualitative analysis offer a replicable model for sub-Saharan African health systems seeking to integrate mental health into PHC services. By incorporating real-world implementation features, such as digital peer learning, structured supervision and role-specific training, the Lagos State MeHPriC program provides evidence-based guidance for policymakers and global mental health stakeholders committed to reducing the mental health treatment gap in Nigeria and similar LMIC contexts.

**Open peer review.**  To view the open peer review materials for this article, please visit http://doi.org/10.1017/gmh.2025.10040.

**Supplementary material.**  The supplementary material for this article can be found at http://doi.org/10.1017/gmh.2025.10040.

**Data availability statement.**  The data supporting the findings of this study are available upon reasonable request from the corresponding author. Due to ethical and privacy considerations, de-identified datasets can be shared with qualified researchers subject to approval from the Lagos State Ministry of Health and the ethics committee.

**Acknowledgements.**  The authors gratefully acknowledge the Lagos State Ministry of Health and the Lagos State Primary Health Care Board for their leadership and support throughout the implementation of the MeHPriC initiative. We are especially thankful to the medical officers, nurses, community health officers, and administrative staff at the 57 Flagship PHCs who participated in the training and evaluation. We also acknowledge the contributions of the training facilitators, supervisors, and qualitative research assistants. Technical advice provided by the Centre for Mental Health Research and Initiative (CEMHRI) is also appreciated.

**Author contribution.**  Abiodun O. Adewuya: Conceptualization, Methodology, Project administration, Supervision, Writing – original draft, Funding acquisition; Bolanle Ola: Investigation, Training delivery, Supervision, Data curation, Writing – review & editing; Olurotimi Coker: Investigation, Resources, Supervision, Validation, Writing – review & editing; Olayinka Atilola: Methodology, Qualitative analysis, Investigation, Writing – original draft, Writing – review & editing; Olushola Olibamoyo: Formal analysis, Supervision, Training coordination,

Writing – review & editing; Olabisi Oladipo: Data curation, Training logistics, Project coordination, Writing – review & editing; Tolu Ajomale: Policy facilitation, Stakeholder engagement, Resources, Writing – review & editing. All authors reviewed and approved the final manuscript and agree to be accountable for all aspects of the work in ensuring its accuracy and integrity.

**Financial support.**  This work was supported by **Grand Challenges Canada** (Toronto, Ontario, Canada) [grant number **TTS-0796-05**] awarded to Abiodun O. Adewuya. Additional in-kind support was provided by the Lagos State Ministry of Health and the Centre for Mental Health Research and Initiative (CEMHRI). No other specific grants from commercial or not-for-profit sectors were received.

**Competing interests.**  The authors declare none.

**Ethics statement.**  Ethical approval for this study was obtained from the Lagos State University Teaching Hospital Health Research Ethics Committee. Written informed consent was obtained from all participants prior to data collection. The study complied with ethical principles for human subjects research, including confidentiality, voluntary participation, and the right to withdraw at any time.

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
