## [Reviewer Report]

Introduction:

The paper did not fully reflect the extent mhGAP integration efforts in Nigeria in their introduction. Authors seem oblivious of a well published earlier and ongoing statewide mhGap integration work in another state. One of the publications of that work, in 2014, titled - “Evaluation of a Mental Health Training Course for Primary Health Care Workers in Ogun State, South West, Nigeria” looked at the effectiveness of PHC workers training on five mhGAP priority mental health conditions. A critical review of the publications on that statewide programme, could have offered the authors a more comparative contextual insight for their work, and provided additional proof of the scientific rigour of this paper.

Priority Conditions:

Anxiety disorders is not one of the named priority mental health conditions in the earlier versions of mhGAP intervention guide, used during this programme implementation in 2017 - it is subsumed under “other significant emotional complaints”, (in 2010), and “other significant mental health complaint” in the 2016 version.

Methodology:

Participants:

The authors explained that the participants composition reflected the diverse roles required for mental health screening diagnosis, management, documentations, referral and follow up. The PHC staff, like pharmacy technicians, counsellors or records/clerical staff, who are not involved in direct patient assessment and treatment process, would require special training modules for their roles, as MHPSS workers, different from the training of frontline clinical cadres. Role guided training content for MHPSS workers is recommended, because PHC Workers should not be trained on roles they are not ethically empowered to perform on the field. This is reflected in the reported result of training effectiveness.

Measurements:

It would be important to state at least the reliability of scales used in the study, which were mainly “adapted” from different sources. Samples of questions may also be presented in the appendix also.

Results:

3-month post intervention assessment:

There was no objective measure of post training mental health service delivery by the participants at the PHC. The self reported practice change survey, and the self-efficacy scale are not objective assessments of service delivery. This was acknowledged as a limitation by the authors. However, a report of 3-months intervention output (e.g, caseloads by diagnostics categories) from the PHC facilties could serve as objective measures of training effectiveness.

Tables:

None of the result tables showed bivariate analysis from which effect sizes were reported.

No “n” is reported for the analysis - This is required to indicate any missing data and how they were managed. This being a prospective study in which some level of attrition is expected.

There was also no report on McNemar’s tests or Cochrane Q which were mentioned in the analysis plan of the quantitative data.

Qualitative data analysis:

The narration of the thematic analysis is inadequate. The 5 reported themes were not well aligned with the 4 classificatory codes derived from the analysis of the qualitative data. The authors may need to map the themes into the CFIR domains codes for better insight into intervention challenges and facilitators. Additionally, the authors may wish to present a thematic map to show interconnections between themes.

Conclusion: This is a well designed and rigorously executed pre-post mixed method study on the impact of WHO mhGAP training on knowledge, attitude, and practices of PHC workers. Attending to the stated observations would further enhance the quality of the paper

---

## [Reviewer Report]

This a well written paper describing one of only a few state-wide attempts to scale up mental health integration into primary care in Nigeria. The authors conducted a thorough mixed method, per-post analysis of data from multidisciplinary teams of primary healthcare workers. The background is well written, methods well described with detailed results and robust discussion.

Minor revisions recommended:

1. I did not see the likert style “10 item self reported practice change survey”. The authors note that they adapted it from two sources, the mhGAP monitoring and evaluation toolkit (2018) and Kirkpartick &Kirkpatrick level three evaluation (2006). However, it will be useful to see what the items are to enable a better understanding of the results.

Same applies to the “self efficacy scale” and “stigma and attitude scale”. They were each adapted from two different questionnaires, but unclear what the items are. Please provide your adapted scale for contextual understanding of the results and for potential reproducibility of your findings.

2. In 3.4 Practice Change at 5 month follow up, the authors state they used a threshold of > 35. Please provide citation to support the choice of this threshold.

3. In page 15, Line 25, The HAPPINESS project... was done in Imo State, Nigeria with is southeastern Nigeria and not northern Nigeria as the authors stated. Please correct

---

## [Reviewer Report]

Thank you for the invitation to review this valuable paper.

I would concur with the statement about the significance of the evaluation in terms of the scale of implementation and evaluation of mhGAP as a resource.

Recruitment/sample

1. There is clearly value in training a diversity of personnel, reaching to non-clinical providers. Was there any difference in the training received, for example by the records/clerical worker, or did they essentially observe, even though the majority of taught content is not directly providing skills they would apply? There is no mention of the Operations Manual, which might be more relevant for some actors in the system.

2. Aside from the definition by technical qualification, was any effort made to assess level of interest/motivation or likely availability to work on mental health in future? In our experience these are essential processes to avoid training large numbers of people who would not use the skills in future.

3. The exact nature of supervision, monitoring and mentoring is important to know, and more detail of exactly what is offered would be valuable for future replication and to understand some of the active ingredients. I note some of these active ingredients (in other studies) are listed in the discussion, but we don’t have a way of seeing how aligned this intervention was to these.

4. It would be helpful to have a brief summary of the policy environment/level of political will in which this work is happening, as this is one factor that can influence success. Similarly major system strengths and weaknesses that may be relevant.

Methods/instruments

These are clear and appropriate.

5. Even if it was not felt necessary to adapt, or the researchers were confident of their validity, some comment on the perceived appropriateness of the instruments for the context would be relevant.

6. I think it would be worth commenting on validity beyond this also, ie to what extent self-reported practice accurately measures actual practice (or what other options for this are available and their relative weaknesses and strengths)

7. While recognising the challenges of reporting mixed methods studies in tight word limits; the qualitative findings are presented only very briefly, and there may be value either in supplementary material or another paper expanding on will have been much more nuanced findings than those presented.

Results

The high proportion of the sample who completed all data is impressive.

8. Do the sample demographic characteristics reflect the make-up of the workforce, or were there any skews (which might bias results)?

Discussion

There is a good level of detail of analysis of the results against other studies, with interesting nuance clearly described. Interesting observation about role-play, which is generally seen as a uniquely valuable element of training in other studies.

9. Again, some reflection on the limitations of instruments in getting to the core of what they are supposed to measure (validity) would be useful. For example, careful language around stigma reduction - ie that the instrument showed a reduction, but social desirability is a likely bias, potentially masking persistence of what are very hard attitudes to change.

Overall a rigorous study and important findings, strengthening the overall case for efficiency of task-sharing interventions as a means of scaling up access to mental health care.

---

## [Editor Report]

Thank you for submitting your manuscript “Scaling Mental Health Care in Nigeria: Impact of WHO mhGAP Training under the MeHPriC Program on Knowledge, Attitudes, and Practices of Primary Health Care Workers in Lagos State – A Pre-Post Mixed-Methods Study” to Cambridge Prisms: Global Mental Health. All reviewers agreed that the manuscript provides a contribution to the value of task sharing in global mental health. However, all reviewers identified concerns with the manuscript in its current form. Kindly attend to the reviewers' concerns and re-submit.